# Exercise-Based Cardiac Rehabilitation Modulates Prefrontal Cortex Oxygenation during Submaximal Exercise Testing in Cardiovascular Disease Patients

**DOI:** 10.3390/bs10060104

**Published:** 2020-06-23

**Authors:** Terence Moriarty, Kelsey Bourbeau, Christine Mermier, Len Kravitz, Ann Gibson, Nicholas Beltz, Omar Negrete, Micah Zuhl

**Affiliations:** 1Department of Kinesiology, University of Northern Iowa, Cedar Falls, IA 50614, USA; kbourbeau@unm.edu; 2Department of Health, Exercise and Sports Sciences, University of New Mexico, Albuquerque, NM 87131, USA; cmermier@unm.edu (C.M.); lkravitz@unm.edu (L.K.); alg@unm.edu (A.G.); zuhl09@unm.edu (M.Z.); 3Department of Kinesiology, University of Wisconsin-Eau Claire, Eau Claire, WI 54702, USA; beltznm@uwec.edu; 4New Heart—Center for Wellness, Fitness and Cardiac Rehabilitation, Albuquerque, NM 87102, USA; omarn@newheartnm.com; 5School of Health Sciences, Central Michigan University, Mount Pleasant, MI 48859, USA

**Keywords:** prefrontal cortex, oxygenation, aerobic exercise, cardiovascular disease, cardiac rehabilitation

## Abstract

The purpose of this study was to investigate if prefrontal cortex (PFC) oxygenation during incremental exercise is altered among cardiovascular disease (CVD) patients who completed 6 weeks of exercise-based cardiac rehabilitation (CR). Nineteen (male = 14, female = 5; 65.5 ± 11.5 years) participants from an outpatient CR program were enrolled in the study. Each participant completed a submaximal graded treadmill evaluation at intake and again upon completion of 18 individualized CR sessions. Functional near-infrared spectroscopy (fNIRS) imaging was used to measure left- and right- PFC (LPFC and RPFC) oxygenation parameters during the submaximal exercise evaluations. Patients showed improvements in cardiorespiratory capacity (pre 5.5 ± 2.5 vs. post 6.9 ± 2.8 metabolic equivalents (METs)). A significant decrease in LPFC and RPFC oxygenation was observed during the post-CR exercise test compared to pre-CR. CVD patients enrolled in 6 weeks of CR showed significant improvements in functional capacity along with decreased cortical oxygenation during submaximal exercise. Exercise training may cause distribution of cortical resources to motor regions that support sustained exercise.

## 1. Introduction

Cortical oxygenation and blood flow have been shown to increase during incremental aerobic exercise among healthy adult humans [1]. Cardiovascular fitness, or exercise training status (trained or untrained), appears to influence the rise in cortical oxygenation and blood flow during exercise. Specifically, adults with higher peak oxygen consumption (VO_2_peak), a measure of cardiorespiratory fitness, demonstrated lower prefrontal cortex (PFC) oxygenation and blood flow than healthy sedentary adults at low, moderate, and hard exercise intensities [1]. The ability to divert blood flow and oxygenation to regions of the brain (e.g., motor cortex) that are required for exercise and away from incidental regions (e.g., PFC) as a result of inhibition of task-irrelevant cognitive processes may help to explain lower cortical oxygenation during exercise among trained individuals [2].

Evidence also suggests that cardiovascular disease (CVD) patients experience augmented PFC blood flow and oxygenation during exercise; however, the rise is diminished compared to healthy controls [3]. This lower rise in PFC blood flow and oxygenation during exercise may lead to a decline in higher order cognitive processing and, for example, impaired gait. It is currently unknown, however, if PFC oxygenation and blood flow changes during exercise are influenced by exercise training status among CVD patients. Cardiac rehabilitation (CR) is a comprehensive program individually tailored to restore CVD patients to optimal functional and vocational status [4]. Aerobic exercise training is a key component of CR, with the primary functional outcome of improved cardiorespiratory fitness. Improvements in VO_2_peak by 3.5 mL·kg^−1^ × min^−1^ (1 metabolic equivalent (MET)) confer reductions in all-cause mortality by 13–34% and CVD events by an estimated 15% [5]. The increase in VO_2_peak as a result of CR may also lead to changes in PFC oxygenation and blood flow during exercise. Therefore, the primary aim of this study is to determine if CVD patients exhibit reduced PFC oxygenation during exercise after training. This will be accomplished by evaluating changes in PFC oxygenation and blood flow during incremental exercise among CVD patients after participation in a 6-week (19.8 ± 1.5 sessions) exercise-based CR program. Cortical oxygenation changes will be monitored via functional near-infrared spectroscopy (fNIRS) imaging technology during a staged submaximal exercise bout performed before and after CR.

## 2. Materials and Methods

Trial Design: The study was a non-randomized single arm trial. Participants who enrolled were assessed before and after 6 weeks of cardiac rehabilitation (within-subject study design).

Patient Eligibility and Study Site: Twenty-eight CVD patients (Table 1) from an outpatient CR program in Albuquerque, NM, volunteered to participate in the study and 19 completed both pre- and post-CR submaximal exercise evaluations. Patients were not excluded based on a history of mood disorders (e.g., major depression), provided that it had been effectively treated (assessed by clinical staff). The University of New Mexico Institutional Review Board approved the study methods (protocol 19218), and all individuals provided written informed consent before participating in this study.

Study Intervention and Outcome Measures: Each patient completed anthropometric and functional capacity estimation exercise evaluations on separate occasions before and after completion of 6 weeks of CR. The fNIRS cap was used to measure PFC oxygenation during the exercise evaluation. Upon enrollment in CR, patients were scheduled for an initial consultation with a cardiologist and clinical exercise physiologist. After intake meetings, patients were fitted with the fNIRS cap and completed a submaximal treadmill exercise test. Post-testing was scheduled during the sixth week of CR. The same procedures were completed at follow-up testing. An outline of the study protocol is shown in Figure 1.

Cardiac Rehabilitation Program: The phase II CR program included individual exercise plans for each patient. They were encouraged to attend three times per week with each exercise session prescribed at an intensity of 50–80% heart rate reserve or rating of perceived exertion of 3–5 (moderate to hard intensity, 0–10 scale) (Borg, 1982) and a duration of 30–60 min. The weekly volume of exercise was progressively increased throughout CR. Attendance at the CR exercise sessions was tracked.

Functional Near Infrared Spectroscopy Recording: PFC oxygenation was measured using a dual wavelength (760 and 850 nm), portable fNIRS system (OctaMon, Artinis Medical Systems, Elst, The Netherlands) during exercise. Four LED optodes (transmitters) combined with one receiver were placed over the right and left PFC regions (RPFC and LPFC; 4 × 2 configuration). Optode placement was based on the modified international electroencephalogram 10–20 system. The left prefrontal cortex (LPFC) regions included Fp1 and F7 using channels 1–4. The right prefrontal cortex (RPFC) included Fp2 and F8 using readings from channels 5–8. The fNIRS cap was placed 2 cm above the nasion and centered on the Fpz location (distinctly depressed area directly between the eyes, just superior to the bridge of the nose) [6]. A source-detector distance of 3.5 cm was used; this is recommended as the optimal distance to detect cortical activity among adults [7,8]. Short separation channels were not implemented in the current study due to potential for introducing error during data analysis [9]. The signal sampling rate was 10 Hz. In order to reduce possible artifact (movement and heart rate), a band-pass filter of 0.01 to 0.50 Hz was applied to the fNIRS signal. Relative concentration changes (∆mol) were measured from resting baseline for oxyhemoglobin (∆O_2_Hb), deoxyhemoglobin (∆HHb), total hemoglobin (∆tHb), and hemoglobin difference (∆Hbdiff). The resting baseline period was established using the first 10 s following a 2-min seated resting period, and defined as 0 mol. This determination was made based on a systematic methodology review of the fNIRS data recording [7]. Oxyhemoglobin change (∆O_2_Hb) is an indicator of PFC oxygenation and is used as an indirect measure of PFC activation [7]. In addition, the change in oxyhemoglobin difference (∆Hbdiff = ∆O_2_Hb − ∆HHb) is considered a sensitive measure of PFC oxygenation due to the high correlation with cerebral blood flow and mean arterial pressure changes [10,11]. Total hemoglobin (∆tHb = ∆O_2_Hb + ∆HHb) change is regarded as a measure of cortical blood flow change. Each fNIRS variable (∆O_2_Hb, ∆HHb, ∆tHb, ∆Hbdiff) for each channel (1–8) were recorded and filtered in Oxysoft software (Artinis Medical Systems, Elst, The Netherlands), and exported into Excel (Microsoft, Redmond, CA, USA). Data was then entered into Prism (GraphPad, San Diego, CA, USA) to complete baseline correction and averaging of each variable and channel across the exercise trials.

Cardiovascular Evaluation: Functional exercise capacity was estimated from a standardized symptom-limited submaximal graded treadmill stress test (Quinton Cardiology). The test began with a 30-s warm-up at 1 mph and 0% incline. Speed was increased each minute by 0.5 mph up to 3.0 mph. Either speed (0.2/0.3 mph) or grade (2%) was increased every minute thereafter. The test was terminated at a rating of perceived exertion of 5 (hard) on the 0–10 scale [12], at the patient’s request, or due to evidence of cardiovascular decompensation or neurologic or musculoskeletal complications. Exercise capacity was estimated in METs at the treadmill test endpoint using final speed and grade [13]. Total walking test time was also recorded in seconds.

Sample Size and Statistical Analysis: The sample size estimation was determined based on a priori calculation for the *t*-test analysis (difference between dependent means) using a power of 0.80 and an alpha level of 0.05 for the selected variables of interest. These included a long-term exercise intervention and changes in PFC oxygenation [14]. The calculated effect size range was 0.6–1.0, which estimated a sample size of 8–18 participants to detect a statistical difference. Twenty-eight were recruited and nineteen completed both pre- and post- testing. All statistical analyses were performed using SPSS Statistics 25.0 (IBM). Paired student’s *t*-tests were used to compare Hbdiff, tHb, O_2_Hb, and HHb changes from baseline (0.00 μmol) within each exercise test using data from the LPFC (fNIRS channels 1–4 averaged) and RPFC (fNIRS channels 5–8 averaged) [15]. The pre-CR exercise test occurred before participation in CR, and the post-CR exercise test was performed after completion of 6 weeks of CR. In addition, paired student’s *t*-tests were used to compare differences before and after 6 weeks of CR for functional capacity values and for PFC measures ∆O_2_Hb, ∆Hbb, ∆Hbdiff, and ∆tHb in the RPFC and LPFC. All results are expressed as means ± standard deviation with a significance level set to a probability value of <0.05 and evaluated for normality. Effect size was established using Cohen’s *d*, where 0.2, 0.5, and 0.8 represent a small, medium, and large effect, respectively [16].

## 3. Results

Patient Characteristics and Changes in Cardiovascular Fitness: Patients displayed improvements in estimated peak functional capacity (6.9 ± 2.8 vs. 5.5 ± 2.5 METs, *d* = 0.53, *p* < 0.01) and average treadmill walk time (470 ± 212 vs. 376 ± 187 s, *d* = 0.47, *p* < 0.01) at completion of CR compared to baseline. No significant changes in body weight or BMI were observed following the 6-week intervention. Participants completed 19.8 ± 1.5 CR sessions over 65 ± 17 days. Participant characteristics are reported in Table 1.

Change in Prefrontal Cortex Oxygenation During Exercise Testing: The RPFC and LPFC ∆O_2_Hb (RPFC: 0.46 ± 0.83 vs. 0.00 μmol, *d* = 0.78, LPFC: 0.60 ± 1.64 vs. 0.00 μmol, *d* = 0.52, *p* < 0.01), ∆tHb (RPFC: 0.65 ± 1.21 vs. 0.00 μmol, *d* = 0.76, LPFC: 0.65 ± 1.72 vs. 0.00 μmol, *d* = 0.53, *p* < 0.05), and ∆Hbdiff (LPFC only: 0.56 ± 2.31 vs. 0.00 μmol, *d* = 0.34, *p* < 0.05) increased from baseline (0.00 μmol) during the pre-CR exercise test. During the post-CR exercise test, PFC oxygenation indicators in both the RPFC and LPFC did not change from baseline. Next, we compared the difference between the pre-CR exercise test and post-CR exercise test for each oxygenation measurement. The RPFC and LPFC ∆tHb were lower during the post-CR exercise test compared to the pre-CR exercise test (RPFC: 0.65 ± 1.21 vs. −0.14 ± 1.33 μmol, *d* = 0.61, LPFC: 0.65 ± 1.72 vs. −0.08 ± 1.54 μmol, *d* = 0.45, *p* < 0.05). In addition, LPFC ∆O_2_Hb was lower at post-CR exercise test compared to pre-CR exercise test (0.60 ± 1.64 vs. 0.09 ± 1.02 μmol, *d* = 0.37, *p* < 0.05) (Figure 2).

## 4. Discussion

The main finding in the current study is that PFC oxygenation was reduced among CVD patients during a submaximal exercise test following 6 weeks of cardiac rehabilitation. Patients in the current study improved cardiorespiratory fitness by an average of 1.4 METs after exercise training; however, cortical activation (measured by O_2_Hb and tHb) in the LPFC and RPFC during exercise was lower following 6 weeks of CR. These data support previous research that reported PFC activity is reduced among older adults during exercise after participation in physical activity programming [14]. Similarly, aerobically trained adults had lower PFC activity during exercise compared to healthy untrained adults. This suggests a link between PFC oxygenation and cardiovascular fitness among CVD patients during a submaximal treadmill walk test.

Activation of the LPFC and RPFC increased during the pre-CR submaximal exercise test as reflected by changes in oxygenation (HbO_2_) and blood flow (tHb). However, the rise was not detected after 6 weeks of exercise-based CR, which resulted in a lower change in the LPFC (via lower HbO_2_ and tHb) and RPFC (via lower tHb) activity during the post-test compared to before CR (Figure 2). Decreased activation or oxygenation in the PFC area indicates lessened use, which may allow cognitive resources to be diverted to other cortical regions (e.g., motor cortex) during walking exercise [14]. Similarly, older adults (mean age 74 years) who participated in 8 weeks of dancing exercise demonstrated lower PFC oxygenation during walking exercise [14]. This neural adaptation to exercise training may be especially beneficial to elderly adults because more oxygenation and activation in motor areas during exercise may possibly support mobility and a reduction in falls. Therefore, it can be speculated that after exercise training there is a shift in metabolic resources and blood flow from the PFC to the motor cortex to focus attention on motor processes during exercise.

An additional explanation for the finding of lower PFC oxygenation after CR is the downregulation of sensory signaling from the skeletal muscle afferents to the central nervous system and, thus, reduced somatosensory input to the PFC, especially during submaximal exercise intensities [17]. With a decreased activation of input to the PFC, individuals may have an added ability to activate other brain regions, such as the motor cortex, and shift resources (i.e., oxygen) to this area after exercise training. Further to this point, many regions of the PFC are involved in the central executive network (e.g., dorsolateral PFC), which requires substantial metabolic resources and is costly to support for a sustained length of time. In this regard, it is possible that some PFC-dependent functions are downregulated to save mental effort and executive resources that are needed to maintain performance during the exercise.

The authors acknowledge the following limitations to the study. Firstly, the small sample size and shorter-term CR program may impact external validity. In addition, cortical activity was assessed in a limited brain region (i.e., the LPFC and RPFC) using a superficial brain tissue measurement during a submaximal treadmill walking test. Therefore, our results may differ from other studies using more invasive measures of cerebral activation (e.g., fMRI), a different exercise modality, or monitoring other brain regions [18]. Other limitations may be due to the interference of skull thickness or the difficulty of predicting how much of an observed signal is due to brain vs. scalp blood flow [19]. Future investigations may benefit from examining data using more global and invasive measures in other brain areas related to walking and their activation adaptations to CR.

The present study demonstrates that there is a reduction in both LPFC and RPFC activation during an incremental submaximal treadmill test following CR in CVD patients. This adaptation suggests that exercise-based CR is effective in freeing up resources otherwise needed for control of locomotion. This may be of practical importance in attention-demanding activities of daily life to prevent injury in this population (e.g., falls at home or car accidents). Future research is warranted to investigate adaptations to other brain areas during exercise as well as to examine the specific duration of time by which a CVD patient can benefit from this important adaptation while living a normal lifestyle outside of and following regimented CR.

## Figures and Tables

**Figure 1 behavsci-10-00104-f001:**
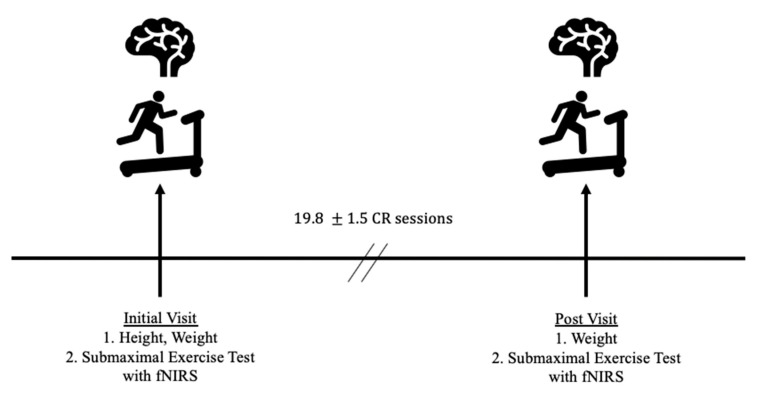
Outline of the study protocol. Number of cardiac rehabilitation (CR) sessions are mean ± SD. fNIRS = functional near infrared spectroscopy.

**Figure 2 behavsci-10-00104-f002:**
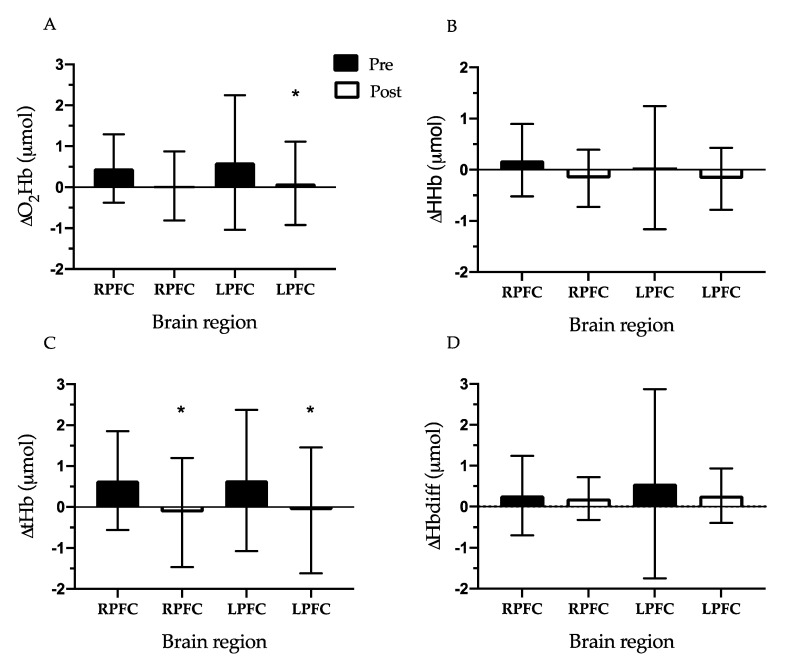
Prefrontal cortex relative changes for functional near infrared spectroscopy measures during the submaximal exercise test pre- and post- cardiac rehabilitation. * significantly lower than pre-cardiac rehabilitation, *p* < 0.05. ∆ = change from baseline (0 μmol), O_2_Hb = oxyhemoglobin (**A**), HHb = deoxyhemoglobin (**B**), tHb = total hemoglobin (**C**), Hbdiff = oxyhemoglobin difference (**D**), LPFC = left prefrontal cortex, RPFC = right prefrontal cortex. Data are mean ± SD, *N* = 19.

**Table 1 behavsci-10-00104-t001:** Baseline patient characteristics (*N* = 28). Mean ± SD or n (%).

Characteristics	Completers	Non-Completers
**Sex, Male/Female**	**14/5**	**8/1**
Age (years)	65.5 ± 11.5	56.8 ± 11.9
Height (cm)	171.0 ± 11.5	171.5 ± 8.8
Weight (kg)	86.7 ± 19.9	92.8 ± 13.5
Body mass index (kg/m^2^)	29.5 ± 6.1	31.7 ± 5.2
Metabolic Equivalents (METs)	5.5 ± 2.5	4.6 ± 2.1
Walking time (s)	376 ± 187	300 ± 157
Coronary Artery Disease	16 (84%)	7 (78%)
Diabetes Mellitus (Type 1 or 2)	5 (26%)	3 (33%)
Congestive Heart Failure	3 (16%)	1 (11%)
Peripheral Vascular Disease	0 (0%)	1 (11%)

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
