# Peer review of "Exercise-Based Cardiac Rehabilitation Modulates Prefrontal Cortex Oxygenation during Submaximal Exercise Testing in Cardiovascular Disease Patients"

_behavsci, 2020, doi:10.3390/bs10060104_

Round 1
Reviewer 1 Report
I would like to commend the authors for this study. It is straightforward and the data is clear.
I address some suggestions / questions to improve the article from my perspective.
How did you summarize NIRS variables registered during exercise (~400s)?
line 92: 'Fp2 and F7. ' I suppose it should be F8.
line 102: 0 micromol. Please, provide rationale for the baseline setup and the definition of 0 micromol.
line 104: I would suggest including the variables and equation for the Hb Diff calculation.
lines 118-120:I would suggest the 'effect size' to show changes from a baseline adjusted to 0 micromol.
line 142: I suppose, Hbdiff = oxyhemoglobin - deoxyhemoglobin
line 148: 'measured by O2Hb and Hbdiff.' I wonder if 'tHb' should be placed here due to the statistical significance.
line 157: 'via lower HbO2.' Shouldn't be lower tHb?
lines 176-182: I would suggest including the study of Fontes et al. Modulation of cortical and subcortical brain areas at low and high exercise intensities. British journal of sports medicine. 2020.
Also, Ferrari et al. Principles, Techniques, and Limitations of Near
Infrared Spectroscopy.Can. J. Appl. Physiol.2004.
Author Response
Reviewer #1: I would like to commend the authors for this study. It is straightforward and the data is clear. I address some suggestions / questions to improve the article from my perspective.
How did you summarize NIRS variables registered during exercise (~400s)?
Response: Thank you for the comment/question. Each fNIRS variable (∆O2Hb, ∆HHb, ∆tHb, ∆Hbdiff) for each channel (1-8) were recorded and filtered in Oxysoft software (Artinis Medical Systems, Netherlands), and exported into Excel (Microsoft, Redmond, CA). Data was then entered into Prisma (GraphPad, San Diego, CA) to complete baseline correction and averaging of each variable and channel across the exercise trials. This has been added to revised manuscript (lines 107-111).
line 92: 'Fp2 and F7. ' I suppose it should be F8.
Response: Thank you for the comment. We have now changed this to reflect the left (Fp1 and F7) and right (Fp2 and F8) PFC. Lines 91-92.
line 102: 0 micromol. Please, provide rationale for the baseline setup and the definition of 0 micromol.
Response: Thank you for the comment. The decision to use this approach for establishing baseline is based on a systematic methodological review analyzing fNIRS data collection procedures (Herold et al 2018, J Clin Med). According to the authors the participants should be in either a seated or supine posture (we used seated). In addition, they cited three studies that used a 2minute resting period, and another 6 studies that implemented a 10s rest. We decided to have the participant sit quietly for 2 minutes, and then use the next 10s after the 2 minutes of rest to establish baseline. A statement has been added to the manuscript (lines 102).
line 104: I would suggest including the variables and equation for the Hb Diff calculation.
Response: We have added the variables and equation for changes in Hbdiff (∆Hbdiff = ∆O2Hb - ∆HHb) and tHb (∆tHb = ∆O2Hb + ∆HHb) (lines 104 & 106).
lines 118-120: I would suggest the 'effect size' to show changes from a baseline adjusted to 0 micromol.
Response: Effect sizes have been calculated for all statistically significant changes and a sentence added to the statistical analysis section in the methods “Effect size was established using Cohen’s d, where 0.2, 0.5, and 0.8 represent a small, medium, and large effects, respectively.” Lines 133-134.
line 142: I suppose, Hbdiff = oxyhemoglobin – deoxyhemoglobin
Response: Yes, this is correct and has been added. Line 106.
line 148: 'measured by O2Hb and Hbdiff.' I wonder if 'tHb' should be placed here due to the statistical significance.
Response: Thank you for the comment. The revision has been made. Line 162.
line 157: 'via lower HbO2.' Shouldn't be lower tHb?
Response: Thank you for catching this. Yes, this has been changed to “via lower tHb”. Line 170-171.
lines 176-182: I would suggest including the study of Fontes et al. Modulation of cortical and subcortical brain areas at low and high exercise intensities. British journal of sports medicine. 2020.
Also, Ferrari et al. Principles, Techniques, and Limitations of Near
Infrared Spectroscopy.Can. J. Appl. Physiol.2004.
Response: Thank you for the comment and recommendations. The above-mentioned articles have been added to the limitations section as references [18] and [19]. Lines 194-196.
Reviewer 2 Report
This work presents a comparison of parameters of prefrontal cortex oxygenation measured before and after six weeks of exercise-based cardiac rehabilitation in a group of 19 patients with cardiovascular disease. The work is interesting and original, and the manuscript is well-written. However, there are several issues that should be addressed, as described below.
Major issues
- Page 2, lines 53 and 54. In the aim of the study, the phrase “…similar to the comparison between aerobically trained versus sedentary healthy adults” suggests that the data from CVD patients were compared against data from healthy adults, which is not the case.
- Page 2, Sub-section “Patients”: About 30% of the initial participants (9 patients) were lost during follow-up. Were the initial characteristics of these lost patients any different than the ones that completed the full study? It is recommended to include a comparison with the characteristics of the lost patients in Table 1.
- Page 4. Statistical analysis.
- Please provide more basic details of the a priori calculation of the sample size (i.e., What test was used and which values were set as reference values?).
- What statistical test was used to verify the normal distribution in the variables?
- Comparisons of prefrontal cortex oxygenation variables include more than one contrast with the same subgroups (i.e., before and after training, left versus right region). Using t-tests for multiple comparisons is not correct unless some correction is applied to all p-values. It is more appropriate to perform analysis of variance for repeated measures (or the equivalent in non-parametric tests in variables where normal distribution was not confirmed by a statistical test).
- Page 4, lines 132 to 133. Why the post-exercise values of O2Hb, tHb, and Hbdiff are expressed as 0.00 and have no standard deviation values? Are these values suppose to be the same depicted in Figure 2?
- Figure 2. The presentation of results as half-bars and one-side error bar is not optimum to represent mean and standard deviation. It is strongly advised to change the representation to dots or circles with error bars in both sides of the mean value, or if the authors want to preserve the presentation of the mean value as a bar, at least make sure to include two-sided error bars (above and below the mean value).
Minor issues
- The title should include the words “cardiovascular disease” instead of “CVD”.
- Page 2, line 63. Regarding the phrase “provided that it had been effectively treated”, How was the depression treatment evaluated?
- Page 4, line 134. The following sentence is unclear: “No measurements changed from baseline during post-CR exercise test”.
- Page 4, line 138. It is recommended to delete the word “see”.
Author Response
Reviewer #2: This work presents a comparison of parameters of prefrontal cortex oxygenation measured before and after six weeks of exercise-based cardiac rehabilitation in a group of 19 patients with cardiovascular disease. The work is interesting and original, and the manuscript is well-written. However, there are several issues that should be addressed, as described below.
Major issues
Page 2, lines 53 and 54. In the aim of the study, the phrase “…similar to the comparison between aerobically trained versus sedentary healthy adults” suggests that the data from CVD patients were compared against data from healthy adults, which is not the case.
Response: Thank you for the comment. This phrase has been removed from the paper.
Page 2, Sub-section “Patients”: About 30% of the initial participants (9 patients) were lost during follow-up. Were the initial characteristics of these lost patients any different than the ones that completed the full study? It is recommended to include a comparison with the characteristics of the lost patients in Table 1.
Response: Thank you for the comment. We have added a “Non-Completer” column to Table 1. In addition, we have decided to rename the “Pre” column to “Completers”, delete the post-CR column and describe the significant changes to functional capacity (in METs and walking time (seconds)) in the results section. Line 137-139.
Page 4. Statistical analysis.
Please provide more basic details of the a priori calculation of the sample size (i.e., What test was used and which values were set as reference values?).
What statistical test was used to verify the normal distribution in the variables?
Comparisons of prefrontal cortex oxygenation variables include more than one contrast with the same subgroups (i.e., before and after training, left versus right region). Using t-tests for multiple comparisons is not correct unless some correction is applied to all p-values. It is more appropriate to perform analysis of variance for repeated measures (or the equivalent in non-parametric tests in variables where normal distribution was not confirmed by a statistical test).
Response: More basic details of the a priori calculation have been added (lines 120-124). The Shapiro Wilk test was used to ensure a normal distribution and a sentence related to normality has been added to the statistical analysis section (line 133). We apologize for the confusion regarding our comparative testing. We performed 2 different sets of paired t-test. First, we compared the changes within each exercise test. We wanted to know if PFC oxygenation changes increased or decreased during the pre-CR exercise test from baseline; and, also if they increased or decreased during the post-CR exercise test from baseline. For this comparison we performed a paired test for each variable (∆O2Hb, ∆HHb, ∆tHb, and ∆Hbdiff) that compared baseline vs. changes in measurements. Second, we performed paired t-tests to evaluate if the changes during the pre-CR exercise test were higher or lower than the changes during the post-CR exercise test. For this comparison we examined the changes across the pre-CR exercise trial to the changes across the post-CR exercise trial for each variable. Therefore, we only made time comparisons (pre vs post), and no regional comparisons (RPFC vs. LPFC). The reason for lack of regional measurements is because we used an 8-channel fNIRS device and separated regions into the RPFC and LPFC. We did not feel confident separating into common demarcations of the PFC. Revisions have been made to the statistical analyses section (lines 126-133).
Page 4, lines 132 to 133. Why the post-exercise values of O2Hb, tHb, and Hbdiff are expressed as 0.00 and have no standard deviation values? Are these values suppose to be the same depicted in Figure 2?
Response: Thank you for the comment and sorry for the confusion. The 0.00 μmol for all variables represents baseline during the pre-submaximal exercise test. This has been clarified by adding a “∆” symbol before each variable which represents a change from baseline as described earlier in the methods. We have also added “(0.00 μmol)” after the word baseline on line 127. No variables changed from baseline during the post-submaximal exercise test.
Figure 2. The presentation of results as half-bars and one-side error bar is not optimum to represent mean and standard deviation. It is strongly advised to change the representation to dots or circles with error bars in both sides of the mean value, or if the authors want to preserve the presentation of the mean value as a bar, at least make sure to include two-sided error bars (above and below the mean value).
Response: Thank you for the comment. Figure 2 has been edited to ensure there are two-sided error bars (above and below the mean value), significance symbols (*) were moved over the post-CR columns for easier viewing and understanding, and the ∆ symbol was added to represent change from baseline. In addition, 0 umol has been added to figure 2.
Minor issues
The title should include the words “cardiovascular disease” instead of “CVD”.
Response: the title has been changed to include the words “cardiovascular disease” instead of “CVD”.
Page 2, line 63. Regarding the phrase “provided that it had been effectively treated”, How was the depression treatment evaluated?
Response: Effective treatment of depression, and determination that the patient was safe to participate was assessed by the clinical staff. This statement has been added to the manuscript (line 62).
Page 4, line 134. The following sentence is unclear: “No measurements changed from baseline during post-CR exercise test”.
Response: The statement has been revised. The statement reflects that no changes in measures of PFC oxygenation changed during the exercise test after cardiac rehabilitation. A further explanation has been provided in the statistical section (lines 128-131) and in the results section (146-147).
Page 4, line 138. It is recommended to delete the word “see”.
Response: the word “see” has been deleted.
Reviewer 3 Report
The authors have made a very interesting research work. The manuscript is well written. I read the manuscript with great interest as changes in cortical oxygenation were monitored through the fNIRS imaging technology during a staged submaximal exercise bout performed before and after cardiac rehabilitation.
I have only a few small suggestions that could, in my humble opinion, improve the manuscript.
-Line 115: Replace “subjects” with “patients (or participants)”
-Lines 131-138: The results must be described in the text or in tables or figures. It is not necessary to repeat them twice.
-Lines 140-143: Since the figures must be independent, i.e. the reader must understand the results without reading the text, I think you could also add the meaning of RPFC and LPFC acronyms. For the same reason, the meaning of fNIRS should be added to line 80.
Author Response
Reviewer #3: The authors have made a very interesting research work. The manuscript is well written. I read the manuscript with great interest as changes in cortical oxygenation were monitored through the fNIRS imaging technology during a staged submaximal exercise bout performed before and after cardiac rehabilitation.
I have only a few small suggestions that could, in my humble opinion, improve the manuscript.
-Line 115: Replace “subjects” with “patients (or participants)”
Response: Thank you for the comment. We have deleted the word “subjects” to agree with another reviewer’s comment.
-Lines 131-138: The results must be described in the text or in tables or figures. It is not necessary to repeat them twice.
Response: Thank you for the comment. Data from table 1 (“Post column”) was removed and the results now described in the text only (results section – lines 137-39). Brain oxygenation results were left in figure 2 and text (lines 143-151) for transparency and if another researcher is attempting to acquire specific values in μmol.
-Lines 140-143: Since the figures must be independent, i.e. the reader must understand the results without reading the text, I think you could also add the meaning of RPFC and LPFC acronyms. For the same reason, the meaning of fNIRS should be added to line 80.
Response: Thank you for the comment. The acronyms noted (RPFC, LPFC, fNIRS) have been added to figures 1 & 2.
Round 2
Reviewer 2 Report
All the issues are solved in the revised manuscript.
Author Response
Thank you again for the thorough review of our manuscript. Your feedback has improved the paper.